psychology

group identity, time perception, religious contexts, internal clock model

**Authors for correspondence:**
Shruti Tewari
e-mail: shrutitewari@iimidr.ac.in
Narayanan Srinivasan
e-mail: nsrini@iitk.ac.in

# Group congruent labelling leads to subjective expansion of time

Shruti Tewari[1], Mukesh Makwana[2,3]
and Narayanan Srinivasan[2,4]

[1]Humanities and Social Sciences, Indian Institute of Management, J207 Academic Block, Rau-Pithampur Road, Indore 453556, India
[2]Centre of Behavioural and Cognitive Sciences, University of Allahabad, Prayagraj 211002, India
[3]Cognitive, Linguistic and Psychological Sciences, Brown University, Providence, RI, USA
[4]Interdisciplinary Program in Cognitive Science, Indian Institute of Technology, Kanpur 208016, India

ST, 0000-0003-1903-7252; MM, 0000-0003-2018-7768; NS, 0000-0001-5342-0381

Given top-down effects on perception, we examined the effect of group identity on time perception. We investigated whether the duration of an ambiguous sound clip is processed differently as a function of group congruent or incongruent source attribution. Group congruent (in-group) and incongruent (out-group) context was created by attributing the source of an identical ambiguous sound clip to Hindu or Muslim festivals. Participants from both the religious groups (Hindus and Muslims) prospectively listened to a 20 s long ambiguous sound clip and reproduced its duration (experiment 1a). Both groups reproduced significantly longer durations when the sound clip was associated with the group congruent compared to the group incongruent festival contexts. The two groups did not differ significantly in reproduced duration when the sound attributed to a non-religious common (busy city street) context (experiment 1b). With multiple durations (1, 5, 10 and 20 s), longer durations were reproduced for group congruent labelling at objectively longer durations (experiment 2). According to the internal clock model of time perception, the significant slope effect indicated that the group congruent context influences temporal experience through changes in pacemaker frequency. We argue that the duration appearing relevant to one's own group is processed differently possibly owing to differences in attentional deployment, which influences the pacemaker frequency.

## 1. Introduction

Groups are essential for human survival. Growing research on social cognition has emphasized the importance of group categorization

on how we make sense of the world and interactions among us. Social categorization influences a variety of perceptual and cognitive processes (see [1]) including face perception [2,3], joint action task [4] and visual processing [5]. Mere sense of in-group membership has profound consequences resulting in biased in-group cognition and interaction (see [6]). The present study extends this line of research to time perception. We investigated whether we perceive the duration of an ambiguous sound clip differently when attributed to group congruent or incongruent contexts in terms of in-group or out-group religious festivals, respectively. We generated an ambiguous sound clip that contained difficult to identify sounds from various religious festivals and busy city streets mixed with white noise.

The perception of time is essential for making sense of the events happening around us. All events are defined over time and its subjective experience shapes our phenomenal experience of events. Eisler [7] has pointed out the underestimation of perceived duration as an archetypical phenomenon. Several models have been used to explain time perception [8,9]. The internal clock model is an influential model proposed to explain the perceived duration. According to this model, duration perception involves multiple stages. Duration perception begins with the *pacemaker* generating pulses at a certain rate, the *switch* allows the pulses to get stored in the *accumulator*. This output is like a memory representation of perceived duration. Then, the *comparator* helps in deciding the length of the perceived duration by comparing the accumulated pulses with the pulses corresponding to the previous reference duration [10–12].

Perceived duration is influenced by various non-temporal factors like saliency [13] emotional content [14] and the amount of the attentional resources allocated to the stimulus [15]. Durations filled with familiar or interesting content are underestimated compared to those filled with unfamiliar content [16]. Perceived time depends on the amount of attention allocated to monitor temporal information; more attention to temporal information leads to longer perceived time, whereas more attention to non-temporal information leads to shorter perceived time [17–21]. Such stimulus-based effects on duration perception are well evident in the time perception research [22].

Interestingly, it is not only the stimulus content, but top-down factors such as the relevance of social context also influence time perception [23–25]. Mere labelling of an identical stimulus (ambiguous sound) in terms of its origin impacts duration estimates [23]. The effect of socio-contextual relevance on time perception was demonstrated in a Hindu festival (Mela) in India. Using a retrospective paradigm, they presented a 20 s ambiguous sound clip to the pilgrims in Mela. They found when the same ambiguous sound clip was attributed to the Mela context, participants reproduced a longer duration compared to when it was attributed to the city context. The effect was observed only when participants knew the source of the sound clip before listening to it. The social meaning associated with the ambiguous sound clip influenced the way information was encoded and in turn affected duration judgement.

In a follow-up experiment using a prospective duration reproduction task, the same effect was replicated [24]. The effect of socio-contextual labelling of stimulus on temporal reproduction disappeared in the dual-task condition, indicating that the observed temporal effect of contextual labelling was probably mediated by attention. In another study, Shankar *et al.* [26] demonstrated that pilgrims volunteered to listen to an ambiguous sound clip for longer time when it was labelled as originating from the Mela rather than from the busy city streets. In addition to this, the ambiguous sound-clip attributed to the Mela was rated as more interesting and less uncomfortable.

These studies demonstrate the importance of social relevance in perceiving time and is consistent with the recent social cognitive research emphasizing the phenomenal experience of 'social' being [1]. However, Srinivasan *et al.* [23] only investigated the effect of the immediate socio-contextual relevance by comparing attribution to pilgrimage and non-pilgrimage context at the pilgrimage site. To our knowledge, no study has investigated the effect of relatively more stable and fixed social belongingness in terms of pre-existing group categories on perceived durations.

The current study aimed to investigate the social labelling effect using intergroup categorization. Classical research on social categories have defined self in terms of shared attributes of a self-inclusive social category (social identity) [27] or collective self-construal based on group membership differentiation in terms of 'us' versus 'them' [28]. Social identity or collective self-construal as in-group and out-group have significant effects on both social vision (for review see [3]) and visual perception [5,29]. Most forms of these intergroup cognitions emerge as the spontaneous and direct outcome of self-categorization into social groups. Social identification emerges early in development and has a pervasive effect on social learning and evaluation [6]. Research on intergroup cognition suggests that intergroup categorization is channelled towards in-group favouritism. These studies demonstrated intergroup differences in basic perceptual processing. Consequences of intergroup categorization are vast and a product of the interaction between low-level perceptual processing and higher-order social cognitive processing [30].

The present research attempts to investigate the effect of intergroup categories (in terms of religious context) on the perceived duration of an identical ambiguous sound clip. We hypothesized mere group congruent labelling would influence duration perception. Using a prospective duration reproduction task, we presented ambiguous sound clip(s) to participants from two religious groups at the study site, Hindus (ethnic majority group) and Muslims (ethnic minority group) by attributing its source to either in-group or out-group festivals. The festivals are celebrated collectively and are highly relevant for respective ethnic groups. Hundreds of people gather on the roads for celebration surrounded by religious and entertaining music played through high volume loudspeakers installed specially for these festivals. Loud noises on streets during these festivals are quite analogous and resembles the cacophony of an everyday busy city street or bus station. Though the level of noise may sound analogous, the subjective meaning people attach to these contexts can be fundamentally different as a function of group membership.

To test the effect of group congruency in terms of in-group and out-group festival-based labelling on time perception, two experiments were carried out. Experiment 1a tested the effect of group congruent labelling on duration perception of a 20 s ambiguous audio clip attributed to in-group or out-group sources using the paradigm used by Srinivasan et al. [23,24]. The same ambiguous stimulus was used in all the conditions to control for low-level stimulus factors and to ensure that the effect is owing to source labelling. Experiment 1b tested whether there is any effect between the two religious groups when the clip was attributed to a common non-religious busy city street context. This experiment was performed to ensure that there is a difference in duration perception between religious groups only when the labelling is based on religious context (in experiment 1a) and the effect disappears when a common secular label (city sounds) is used. experiment 2 focused on replicating and testing the group congruency effect with multiple durations; viz. 1, 5, 10 and 20 s to understand the mechanisms involved in the in-group relevant context-related temporal effect in terms of a clock model of time perception [10–12]. We picked these durations (lasting 1–20 s) because the processing of longer durations involves more cognitive resources [31] that could potentially be influenced by group congruent labelling and in turn influence mechanisms involved in time perception.

# 2. Experiment 1a

To test whether group membership influences perceived duration, we presented a 20 s long identical ambiguous sound clip containing very little recognizable content to participants from two ethnic groups (Hindus and Muslims). The source of the sound clip was attributed to either Hindu festivals or Muslim festivals, making the source congruent or incongruent to group membership (Hindus and Muslims). We predicted longer duration reproduction when the source attribution was congruent to group (religious) identity, i.e. in-group context, compared to when it was incongruent, i.e. out-group context.

## 2.1. Method

### 2.1.1. Participants

Ninety-two male student volunteers (mean age = 21.49 years, s.d. = 2.43 years) from Allahabad University participated in the study after filling in a written consent form. We restricted our recruitment only to male participants because of the limited availability of female participants in a particular condition. Participants self-reported themselves as either Hindus ($n = 46$, mean age = 21.46 years, s.d. = 1.76 years) or Muslims ($n = 46$, mean age = 21.52 years, s.d. = 2.99 years).

A power analysis and a priori sample estimation was performed based on the effect size from Srinivasan et al. [24]. The effect size (Cohen's d) from Srinivasan et al. [24] was 0.8 for the social context effect. For a between-subjects ANOVA with two factors and two levels in each factor, we expected a main effect of religious context with an effect size of $f = 0.4$ (given $d = 0.8$ from the prior study). With $f = 0.4$, $\alpha = 0.05$ and power = 0.95, we calculated the required sample size of 84. However, expecting some exclusions, we recruited 92 participants.

All the experiments followed the guidelines approved by the Institutional Ethics Review Board of University of Allahabad. All participants were compensated with 50 INR for participation.

### 2.1.2. Stimuli

We prepared an ambiguous sound clip that contained sounds recorded from loud festival sites, both Hindu (i.e. Dussehra and Diwali) and Muslim (i.e. Eid and Moharam) as well as from equivalent noisy

city sites such as busy streets, bus stops and market places. These sounds were jumbled together with white noise to make the sounds unidentifiable. The duration of the sound clip was 20 s and the loudness was 85 dB. The 20 s sound clip is available at https://osf.io/g82jf/.

### 2.1.3. Procedure

We employed a prospective temporal reproduction paradigm in which the participants were informed that they would be asked to judge the duration of the stimulus. Participants listened to the 20 s long audio clip through headphones that blocked external noise. Immediately after listening to the sound clip, participants reproduced its duration using a stopwatch. The display of the stopwatch was neither visible to the experimenter nor the participant. The participant pressed a 'start' button, and then, when they felt that the duration for which they originally listened to the clip had elapsed, they pressed the 'stop' button on the stopwatch. This duration was recorded as the reproduced duration. Participants were asked to take off their watches and there were no other devices in the laboratory that could facilitate accurate duration estimation. To avoid counting or any other possibility of time estimation, we instructed participants not to count which has been shown to be a good strategy to minimize or control estimations by counting [32]. The participants were also given explicit instructions as, '*it is not a test of accurate duration estimation of the sound clip rather their task is to report how long the sound clip appears to them*'. Before listening to the experimental stimulus, participants received a practice trial with a 5 s long unrelated sound clip (to familiarize them with the duration reproduction task).

Participants were randomly assigned to one of the two labelling conditions. Half of the participants from both the ethnic groups, were told that the audio clip was recorded from the Hindu festivals and the other half were told that the sound clip was recorded from Muslim festivals. The experiments consisted of a single trial and the identical 20 s long audio clip was presented only once in each of the conditions. This instruction was given verbally in Hindi language, and English translation as follows:

> Hindu Festival condition: '*we have jumbled together various sounds recorded from different Hindu festivals —for example, Dussehra, Diwali etc. OK? So, now you are going to listen to the sounds recorded from the Hindu Festivals.*'

> Muslim Festival condition: '*we have jumbled together various sounds recorded from different Muslim festivals—for example, Eid, Moharram, etc. OK? So, now you are going to listen to the sounds recorded from the Muslim Festivals.*'

To check whether participants processed the information about the source of the sound and remembered it until the end, all participants were asked to recall the source of the sound clip at the end of the experiment.

## 2.2. Results

Eight participants (three from the Hindu festivals condition and five from the Muslim festivals condition) were excluded because they failed to correctly recall the source of the sound clip after duration reproduction (*n*: Hindu in-group context = 23; Hindu out-group context = 20; Muslim in-group context = 20; Muslim out-group context = 21).

We performed a 2 (religion: Hindu and Muslim) × 2 (source attribution: Hindu and Muslim festivals conditions) between subject ANOVA on reproduced durations. The main effect of religion was significant, $F_{1,80} = 8.87$, $p = 0.004$, $\eta_p^2 = 0.10$. Muslim participants gave longer estimates compared to Hindu participants ($M_{\text{Hindu}} = 10.32$, 95% confidence interval (CI) (9 s, 11.63 s) and $M_{\text{Muslim}} = 13.14$, 95% CI (11.79 s, 14.49 s)). The main effect of source attribution was not significant $F_{1,80} = 0.26$, $p = 0.61$, $\eta_p^2 = 0.003$ ($M_{\text{Hindu festivals}} = 11.97$, 95% CI (10.67 s, 13.27 s) and $M_{\text{Muslim festivals}} = 11.48$, 95% CI (10.13 s, 12.85 s)).

More importantly, the interaction between religion and priming condition was significant, $F_{1,80} = 18.18$, $p < 0.001$, $\eta_p^2 = 0.19$. *Post hoc* comparisons (Holm–Bonferroni corrected) showed both Hindu and Muslim participants gave longer duration estimates when primed with in-group labels. The Hindu participants reproduced longer durations for Hindu festivals ($M = 12.58$, 95% CI (10.78 s, 14.38 s)) compared to Muslim festivals ($M = 8.06$, 95% CI (6.13 s, 9.99 s)), $t_{42} = 3.41$, $p = 0.005$, $d = 1.32$. The Muslim participants reproduced longer durations for Muslim festivals ($M = 14.92$, 95% CI (12.99 s, 16.85 s)) compared to Hindu festivals ($M = 11.36$, 95% CI (9.48 s, 13.25 s)), $t_{39} = 2.63$, $p = 0.04$, $d = 0.69$. Hindu participants produced ($M = 8.06$, 95% CI (6.13 s, 9.99 s)) shorter duration compared to Muslim participants ($M = 14.92$, 95% CI (12.99 s, 16.85 s)) when primed with Muslim festivals, $t_{42} = 5.0$, $p < 0.001$, $d = 1.42$. However, both the religious groups gave similar duration estimates, when primed with Hindu festivals, $t_{42} = 0.93$, $p = 0.36$, $d = 0.32$ (figure 1).

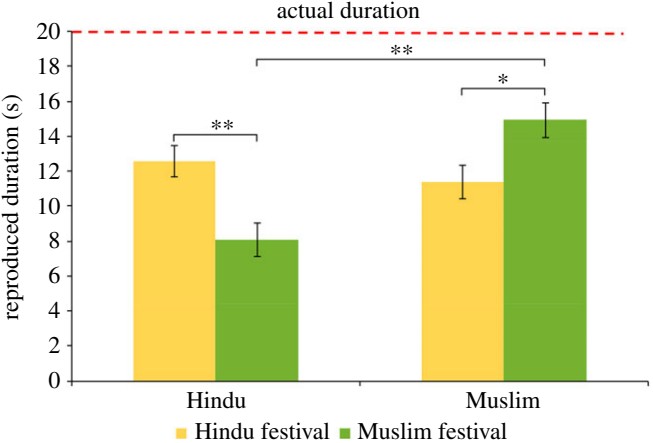

**Figure 1.** Reproduced duration as a function of participants' religious group (Hindu or Muslim) and sound's attributed origin (from Hindu or Muslim festivals).

# 3. Experiment 1b

As a control condition, we also assessed duration estimation of the identical 20 s ambiguous sound clip across both the religious groups attributing it to a common, non-religious context; the busy city street. We hypothesized that this is a common social context without any relevance to specific groups, there should be no difference in the reproduced duration between the two groups.

## 3.1. Method

### 3.1.1. Participants

Forty-three male students (mean age = 21.86 years, s.d. = 2.49 years) from Allahabad University voluntarily participated in the study after filling in a written consent form. They self-reported themselves as either Hindus ($n$ = 21, mean age = 21.10 years, s.d. = 1.51 years) or Muslims ($n$ = 22, mean age = 22.59 years, s.d. = 3.02 years).

### 3.1.2. Procedure

We employed a prospective paradigm (as in experiment 1) in which the participants were informed that they would be asked to judge the duration of the stimulus. Participants listened to the 20 s audio clip through headphones that blocked external noise. Immediately after listening to the sound clip, participants reproduced its duration using a blind folded stopwatch. The method was the same as in experiment 1. Participants were told that the audio clip was recorded from the marketplace of a regular busy city street. The instructions were delivered in Hindi and English versions as follows:

> City condition: 'we have jumbled together various sounds from the city—for example, from markets, railway stations, bus station, various places in the city. OK? So now you are going to listen to the sounds of the city.'

## 3.2. Results

To check whether participants processed the information about the sound's source, all participants were asked to recall the source of the sound clip at the end of the experiment. Two participants (one from each of the ethnic groups) were excluded owing to erroneous recall of the sound's source ($n$: Hindus = 20 and Muslims = 21). Independent samples $t$-tests on reproduced durations did not show a significant effect, $t_{40}$ = −0.73, $p$ = 0.47, $d$ = 0.23, 95% CI: (−3.37, 1.58). Both the groups reproduced similar durations when the source of the sound clip was labelled as neutral city context ($M_{Hindu}$ = 12.86 s, s.e. = 0.90, $M_{Muslim}$ = 13.76 s, s.e. = 0.84). We also calculated the Jeffrey–Zellner–Siow prior Bayes factor and obtained 2.64, indicating evidence for the null hypothesis.

To summarize experiments 1a and 1b, we found significant effects as predicted with both the groups reproducing longer duration estimates when the source of the sound clip was labelled with in-group context compared to out-group context in experiment 1a and a null effect with a common social

context in experiment 1b. The results indicate that the perceived duration of an identical ambiguous sound varies as a function of its group congruent and incongruent labelling.

# 4. Experiment 2

The second experiment investigated the mechanism underlying the group congruent source labelling effect in experiment 1a. In the context of the internal clock model [10], the slope refers to the rate at which the pacemaker encodes the temporal representation, whereas the intercept refers to a constant shift in the temporal representation independent of the pacemaker and mostly attributed to the switch component. Change in slope implies that the pacemaker component of the model has been possibly influenced, whereas changes in intercept (latency) implies that probably the switch component of the clock model has been affected.

Applying this to the group congruent source labelling effect in experiment 1, if group congruent/ incongruent labelling leads to changes in the pacemaker component of the internal clock, then the changes in temporal judgements would increase in a multiplicative manner with an increase in the magnitude of actual duration, leading to a slope effect [10,12]. On the other hand, if the source labelling leads to changes in the latency of the switch component, then the changes in temporal judgements would increase in an additive manner with an increase in the magnitude of actual duration, leading to an intercept effect.

To examine the underlying process using the clock model, we presented a series of ambiguous sound clips of four durations, i.e. 1, 5, 10 and 20 s. We attributed their sources to either in-group or out-group festivals and asked the participants to reproduce duration estimates by pressing the space bar. As the primary intent was to test the effect of group congruent labelling on duration estimation of four durations and identify mechanisms influenced by group congruency in the framework of the clock model [12], we used only one ethnic group; the Hindu group and included participants from both genders.

## 4.1. Method

### 4.1.1. Participants

Forty-seven students (26 males, mean age = 21.08 years, s.d. = 1.87 years) from Allahabad University gave written consent to participate in the study. All the participants self-reported themselves as Hindus.

### 4.1.2. Stimuli and procedure

Participants performed a prospective duration reproduction task in a computer-based experiment using EPRIME2 [33].The 20 s ambiguous sound clip (experiment 1) was randomly edited into different clips of four durations; 1, 5, 10 and 20 s. Each sound clip of a specific duration was repeatedly presented for five trials. The order of presentation of the sound clips was randomized. Immediately after listening to each clip, participants reproduced the duration by pressing and holding the spacebar key. Half the participants were instructed that sounds were recorded from in-group (Hindu festivals) context and other half were instructed out-group (Muslim festivals) source attribution. For manipulation check of the labelling of the sound origin, participants were asked to describe the source of the sound clip at the end of the experiment. To ensure the groups are similar in terms of religious factors, the participants were asked few questions on the level of religiosity (to what extent you consider yourself as religious?), Hindu identification (to what extend you identify yourself as Hindu?), contact with Hindus (how many Hindu friends do you have?/how many Hindus live close to your residence?/ how often do you spend time with Hindu friends?) and contact with Muslims (how many Muslim friends do you have?/how many Muslim live close to your residence?/how often do you spend time with Muslim friends?) using five point rating scales (ranging from not at all to a lot) along with demographic details such as age, gender and education.

## 4.2. Results

Eight participants (five from the Muslim festivals condition and three from the Hindu festival condition) were excluded as they erroneously recollected the sound's source (n: Hindu prime = 18; Muslim prime = 21).

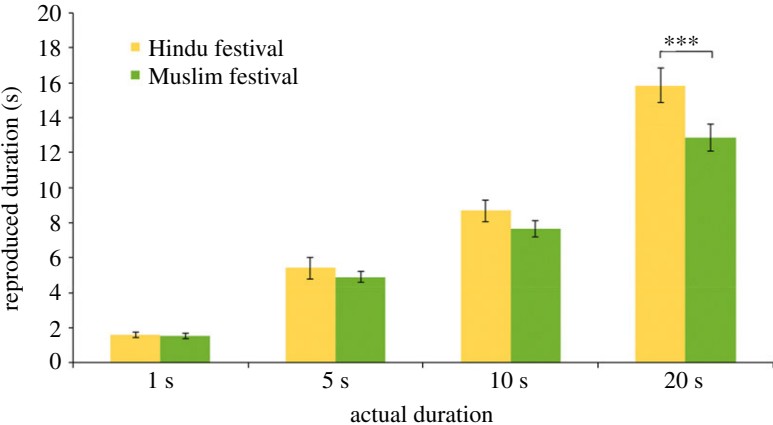

**Figure 2.** Reproduced estimates of four durations across sound's source attributions (from Hindu or Muslim festivals).

We performed $t$-tests to check whether the two groups differed in terms of religiosity, identity and contact with in-group and out-group members. The difference between religiosity for those in the Hindu condition ($M = 3.06$, s.d. $= 0.25$) was not significantly different compared to those in the Muslim condition ($M = 2.86$, s.d. $= 0.26$), $t_{37} = 0.55$, $p = 0.59$ $d = 0.18$. Similarly, the difference between Hindu identity across conditions Hindu ($M = 3.56$, s.d. $= 1.25$) and Muslim condition ($M = 3.62$, s.d. $= 1.20$) was not significant, $t_{37} = 0.16$, $p = 0.87$, $d = 0.05$. The two groups were also similar with respect to the rating of everyday contact with Hindus and Muslims. Mean contact ratings with Hindus did not differ significantly between those in the Hindu condition ($M = 4.74$, s.d. $= 0.39$) compared to those assigned to the Muslim condition ($M = 4.84$, s.d. $= 0.27$), $t_{37} = 0.92$, $p = 0.36$, $d = 0.30$. Similarly groups had similar amount of contact with Muslims across the Hindu condition ($M = 2.94$, s.d. $= 0.87$) and Muslim condition ($M = 2.73$, s.d. $= 0.65$), $t_{37} = 0.88$, $p = 0.38$, $d = 0.28$.

To test the effect of group congruent source labelling, we performed a 4 (duration: 1, 5, 10 and 20 s) × 2 (source attribution: Hindu and Muslim festivals) mixed group ANCOVA on reproduced durations with religiosity, Hindu identification, contact with Hindu and contact with Muslims as covariates. Greenhouse–Geisser correction was applied as sphericity was found to be violated by Mauchly's test of sphericity. Significant main effect of context showed longer duration estimates in the Hindu festivals condition ($M = 7.89$, 95% CI: 6.97 s, 8.81 s) compared to the Muslim condition ($M = 6.62$, 95% CI: 5.77 s, 7.47 s), $F_{1,33} = 7.59$, $p = 0.009$, $\eta_p^2 = 0.19$. We successfully replicated the group congruent labelling effect that was established in the first experiment. The main effect of duration was also significant: $F_{1.93,63.75} = 10.00$, $p < 0.001$, $\eta_p^2 = 0.23$, showing that all the estimated durations differed significantly from each other (figure 2). This shows that the participants were able to perform the task and discriminate different durations.

More importantly, the interaction between duration and group congruent/incongruent source attribution was significant: $F_{1.93,63.75} = 7.56$, $p = 0.001$, $\eta_p^2 = 0.19$. *Post hoc* comparisons (Holm–Bonferroni correction) showed longer reproduction for the in-group ($M = 15.86$, 95% CI: 14.06 s, 17.66 s) as compared to out-group condition ($M = 12.67$, 95% CI: 11 s, 14.33 s) only for the 20 s duration: $t_{37} = 4.90$, $p = 0.001$, $d = 0.85$. A similar trend was seen in other durations such as 5 s ($M_{\text{Hindu}} = 5.42$, 95% CI: 4.08 s, 6.76 s; $M_{\text{Muslim}} = 4.70$, 95% CI: 4.05 s, 5.34 s, $t_{37} = 1.46$, $p = 0.30$, $d = 0.34$) and 10 s ($M_{\text{Hindu}} = 8.75$, 95% CI: 7.64 s, 9.87 s; $M_{\text{Muslim}} = 7.62$, 95% CI: 6.62 s, 8.63 s, $t_{37} = 2.04$, $p = 0.13$, $d = 0.47$) though the difference was not significant (figure 2).

None of the interactions with covariates except between religiosity and durations were significant, $F_{1.93,63.75} = 6.09$, $p = 0.004$, $\eta_p^2 = 0.16$. This effect of religiosity and duration was independent from the interaction between group congruent source attribution and duration. Further analysis showed that religiosity influences duration estimation at larger durations of 10 and 20 s ($ps < 0.05$) but does not influence duration estimations of 1 and 5 s ($ps > 0.05$).

To analyse whether in-group labelling leads to differences in slope or intercept, for each participant, we fitted a linear regression line for reproduced duration as a function of actual duration. The results showed that the slope values were significantly higher in the in-group condition ($M = 0.74$, 95% CI: 0.64, 0.83) compared to out-group condition ($M = 0.58$, 95% CI: 0.50, 0.64): $t_{37} = 2.69$, $p = 0.011$, $d = 0.86$. However, there was no significant difference in the intercept values between the two conditions ($M_{\text{Hindu}} = 1.27$, 95% CI: 0.46, 2.07 and $M_{\text{Muslim}} = 1.45$, 95% CI: 1.10, 1.80): $t_{37} = 0.44$, $p = 0.66$, $d = 0.14$.

The results confirmed our hypotheses and indicate the group congruent or incongruent labelling effect (in-group versus out-group) influences the pacemaker component in the internal clock model.

# 5. General discussion

The current findings confirmed and replicated the impact of group identity based on social labelling of duration estimation of ambiguous auditory stimuli. Participants reproduced longer durations when source of the ambiguous sound clip was labelled as being associated with a group congruent context, i.e. in-group festivals compared to group incongruent context, i.e. out-group festivals. Our research contributes to the burgeoning literature demonstrating top-down processing effects on perception. Keeping participants' personal attributes and the stimulus constant, the present findings showed merely attributing a sound clip to group congruent or incongruent sources affects its duration estimation.

Results clearly showed that both the ethnic groups reproduced longer estimates for the sound clip attributed to the in-group sources compared to the out-group source. While all the source attributions (in-group and out-group festivals) have social meanings attached to it, the in-group attribution may make the sound clip to appear semantically rich, more relevant, fluent and familiar for absorbing attention. On the other hand, the out-group attribution may limit deployment of the attention owing to unfamiliarity resulting in shorter estimates. Using dual task in a similar prospective duration experiment, previous research has demonstrated the mediating role of attention in the effect of socio-contextual labelling on reproduced duration [24]. The null effect in reproduced duration in the case of the common context (labelling of auditory clip as from city streets) supplement the robust effect of group relevant social meaning attached to the stimuli.

The effect of group categorization is pervasive; with significant effects on attitudes [6,34], attribution [35] memory [36] and face perception [3]. Growing research indicates the neurological basis of unique in-group cognition [37,38]. However, no study to the best of our knowledge has so far studied the direct effect of group relevant labelling in reference to in-group and out-group categorization on duration estimation.

A significant main effect of religion on duration estimates (experiment 1a) can be explained as an asymmetrical effect owing to the social status of the ethnic groups. Research has highlighted that people are naturally motivated to develop rich, elaborated and differentiated cognitive structures to represent information related to their in-group rather than to other groups [39]. These group effects are asymmetrical based on relative social status, power and outcome dependency [40]. Thus, being a member of minority or majority ethnic groups may affect the way one processes group relevant information. People in minority and subordinate positions have regular first-hand interactions with the majority owing to mutual goals and needs. It makes the out-group more familiar and socially relevant [41]. This could be a possible reason for overall longer duration estimates by the Muslim group (ethnic minority group) as compared to the Hindu group (ethnic majority group).

Given that the audio clip was identical and ambiguous across all conditions in experiment 1a, the temporal processing changed owing to mere in-group or out-group labelling and not owing due to purely bottom-up factors. The two religious groups did not show any difference in duration perception when they were presented with a common secular context (experiment 1b). The results of experiment 2 further confirmed that purely bottom-up sensory processing differences cannot account for our group congruent labelling results. The participants interpreted the in-group festival sounds possibly as having meaningful associations resulting in the richer encoding of (ambiguous) stimuli. This meaningful association to in-group festivals probably led to top-down influences that include motivational processes and attentional processes on mechanisms involved in time perception. The role of motivational and/or attentional mechanisms underlying the effects of intergroup categorization on duration estimation need further exploration. The results indicate that understanding how we perceive time needs an integrative approach to understand the joint effects of non-temporal perceptual attributes of sensory stimuli, the allocation of attentional processes, and the existence of the previously established mental representations related to our categorizations [22].

The robustness of this group labelling effect was replicated and extended to both males and females using key press reproduction method for 20 s sound clips. We found a significant slope effect suggesting that group labelling may influence time perception through changes in the pacemaker speed. Srinivasan *et al.* [24] explained that a similar effect on social meaning for duration estimation of a 20 s long audio clip is probably mediated by attentional mechanisms. While the attentional gate model has argued that attention mainly is associated with the opening and closing of the switch component, the results of the current study indicate that social context may increase the pacemaker speed [12]. Longer duration

possibly enables more attention to social content present in the relevant auditory clip along with other noisy information. Further studies are needed to elucidate how group congruency influences duration estimation using other models of time perception [42].

Deeper understanding of these intergroup effects on duration estimation has important implications. Research on intergroup biases often demands motivational and cognitive explanations. Empirical evidence supporting how mere group labelling affects duration estimation complements this line of research. Time perception shapes the subjective experience of the events and may have varied consequences depending on which group one belongs to.

These experimental findings add value to cognitive psychology, time perception and social categorization literature. We commenced these studies to better understand the effects of top-down social factors on basic cognitive processes. The empirical evidence clearly demonstrates a state-level effect of religious labelling on time perception. The significant association between trait level religiosity and time perception warrants a systematic exploration and explanation in future studies. These studies also showcase an experimental approach to study the effect of intergroup categorization. The social context effect on time perception has already been shown in Mela context [23]. Our findings reveal how pre-existing social identity in terms of religious groups affect one's subjective experience of time. The group congruent context influences temporal experience possibly owing to differences in attentional deployment associated with identity-salient stimuli. Experimental studies can further illuminate the theoretical basis of such in-group effects and refine the understanding of group biases in perception, in general, and more specifically time perception. Future explorations are needed to explore how changes in time perception owing to intergroup categorization may influence downstream behavioural biases such as risk-taking behaviour, political choices and economic decisions, which depend on temporal factors.

Ethics. All the experiments followed the guidelines approved by the Institutional Ethics Review Board of the University of Allahabad

Data accessibility. We confirm that all data required are available from https://doi.org/10.17605/OSF.IO/XTQ84 [43].

Authors' contributions. S.T. and N.S. came up with the idea and planned the study. S.T. and M.M. designed experiments, collected data and performed analysis. S.T. initially drafted the manuscript. N.S. and M.M. critically revised the manuscript. All authors approved the final version of the manuscript for submission.

Competing interests. The authors declared that there were no conflicts of interest with respect to the authorship or the publication of this article. At the time of writing, Prof. Narayanan Srinivasan is a Board Member of the Royal Society Open Science, but had no involvement in the review or assessment of the paper.

Funding. The authors have received no funding for this research.

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
