## [Reviewer comments · Royal Society Open Science]

Review History

RSOS-201063.R0 (Original submission)

Review form: Reviewer 1

Is the manuscript scientifically sound in its present form?

Yes

Are the interpretations and conclusions justified by the results?

Yes

Is the language acceptable?

Yes

Do you have any ethical concerns with this paper?

No

Have you any concerns about statistical analyses in this paper?

No

Recommendation?

Accept with minor revision (please list in comments)

Comments to the Author(s)

The strong point of this manuscript is the systematic variation of the social / contextual embedding of the stimuli to be judged in duration. The same ambiguous stimuli were used with different contextual attributions. Scrambled noise created from recorded street scenes and religious festivals were verbally attributed as stemming from either Hindu or Muslim festivals. The stimuli are ambiguous because the noise is always the same, only the verbal attribution differs. Hindu and Moslem students listened to the sound clips and relatively over-reproduced the sounds of their own (in-group) religious festivals as compared to the other (out-group) religious festival. In a control experiment where the noise was attributed to normal street noise the religious groups did not differ in their reproduction performance. The study seems simple enough but unfolds with different parts to create meaningful complexity. The reproducing is done with only a long (20 s stimulus), with a variation of four different durations (between 1 and 20 s), with the in-group and out-group attribution, or a neutral attribution and an equal share of Muslim and Hindu participants. What is unique here is that participants did not fulfil typical task requirements and stimulus properties which would be similar to everyone (a sinus tone, or even emotionally charged stimuli), but that information on the noise which would trigger an individual group background, a contextual (top-down) embedding significantly influenced the duration reproduction performance in otherwise always identical stimuli.

Here are a few remarks on details in the manuscript:

- When on page 4 the term “ambiguous sound clip” appeared, I did not fully understand what that meant. I had to wait until I was reading the methods section to fully understand it. Since this is an important term in the context of this study I would early on give a fuller explanation so that one can better follow the introduction.
- It is probably not so important but one takes notice and asks why female students were unavailable (page 7) in experiment 1a.
- The power analysis on page 7 leads to $n = 35$ subjects for a two-group comparison. However the decisive interaction effect contains $n = 20$ members of the subgroups.
- In the Results section for all experiments there is the outlier criteria of subjects “erroneously recollecting the source of the sound”. Within the given context I don’t understand this. The participants are given the information of Muslim, Hindu, or neutral context. What is an erroneous recollection of sounds?
- Page 17: Make explicit which group is the ethnic minority in Allahabad, I presume the Muslim community?

Review form: Reviewer 2

Is the manuscript scientifically sound in its present form?

No

Are the interpretations and conclusions justified by the results?

No

Is the language acceptable?

Yes

Do you have any ethical concerns with this paper?

No

Have you any concerns about statistical analyses in this paper?

No

Recommendation?

Major revision is needed (please make suggestions in comments)

Comments to the Author(s)

The study deals with an interesting phenomenon – subjective passage of time related to religious content/attribution in the situation. Own-group attribution is demonstrated to be related to longer time productions of the audio clip. Unfortunately the study remains too superficial and descriptive, not aiming to explain the reported effect. Actually, it also explains but the explanation isn't totally convincing (as simpler causes of the reported effect aren't excluded, see below). They also don't use entirely all means experimental psychology has developed to get closer to the causes of the effect (e.g., subjective reports on how the participants had encoded/reproduced the clips, gradual religiosity related effect of subjective time, within-participants design etc).

Altogether, introduction and experiments themselves doesn't separate social context from other possibly important aspects in the situations (like emotional meaning, familiarity of the situation/festivals, direction of attention etc) that may capture encoding, perception and emotions, and thus also have an effect on time perception. Used terms like "belongingness" are not very well-defined and may be unnecessarily fancy (especially if we agree that these confusions are there). Please explain these issues more deeply!

Some important experimental details are missing (how many repetitions? how many different clips were used? what is the objective description of them, i.e., how many important details can be read out from there? how was the religiousness of participants taken into account? did the groups differ? what was the "recalling of the source of the sound clip, p10, lines 17-23).

How to separate social context from emotions and other factors? Does top-down influence mean that low-level explanations are ruled out? I don't think so (as top-down knowledge may easily direct ones's attention to low-level features in the event and thus exclude or minimize high-level effects; the phenomena like perceptual learning, different memory content or higher perceptual sensitivity may play a role there, that in turn are important in time perception models cited above , p 4 of 22)

Please explain what slope and intercept (p 16 of 22) mean in the context of time perception?

It's still possible that Exp 1a and 1b reflect individual differences, not only experimental manipulation. I consider it as a weakness that between-participants not within-participants design was used (Exp1a and 1b , Exp2)

Memory content effect and direction of attention effects (relevant to time-perception theories referred to in Introduction) may be relevant here, but aren't studied more.

Altogether, I suggest major revision so that (1) simpler factors/explanations would be considered; (2) mechanism how religious belongingness could affect time perception (i.e., relate it to used models of time perception); (3) give experimental details, incl independent description of used sound clips (what do they possibly contain for the groups); (4) describe your groups more across dimensions you consider important in your study (religiosity, belongingness, memory span, memory content in the experiment etc) to rule out other and support your conclusions

Decision letter (RSOS-201063.R0)

Dear Dr Tewari

The Editors assigned to your paper RSOS-201063 "Group congruent labelling leads to subjective expansion of time" have now received comments from reviewers and would like you to revise the paper in accordance with the reviewer comments and any comments from the Editors. Please note this decision does not guarantee eventual acceptance.

Please submit your revised manuscript and required files (see below) no later than 21 days from today's (ie 10-Aug-2020) date. Note: the ScholarOne system will 'lock' if submission of the revision is attempted 21 or more days after the deadline. If you do not think you will be able to meet this deadline please contact the editorial office immediately.

on behalf of Dr Alexa Morcom (Associate Editor) and Essi Viding (Subject Editor)
openscience@royalsociety.org

Associate Editor Comments to Author (Dr Alexa Morcom):

Associate Editor: 1

Comments to the Author:

Dear Dr Tewari

Thank you for submitting your manuscript "Group congruent labelling leads to subjective expansion of time" (RSOS-201063) to Royal Society Open Science. We have now received two

reviews. Although the reviewers feel that these results are of potential interest, they differ in their evaluation of the paper's potential significance, and have raised some substantive concerns. These issues preclude publication of the paper in Royal Society Open Science, at least in its present form.

Reviewer 2 questions the theoretical framing of the experiments, and asks that you consider simpler explanations that might underpin the effects of the social context manipulation.

Reviewer 1 also asks you to give further detail about the power calculations used to justify the sample sizes. It is essential to justify the samples for the specific effects of primary interest. In addition to the reviewers' points, I note that in Experiment 1a, the p value for the post-hoc test for the Muslim groups on p.11 lines 19-20 is quoted as equal to 05. Is this correct, or is it less than .05? At $\alpha = .05$, this distinction is important as it affects whether this comparison can be declared significant or not, and thus the interpretation of the interaction.

The two reviews are appended to this message. We hope that you will be able to address the reviewers' concerns in full and resubmit the manuscript, along with a point-by-point reply to the reviews that indicates your response to each concern. Before we make a decision about publication, we will have your revision re-reviewed.

Best wishes
Alexa

Associate Editor: 2
Comments to the Author:
(There are no comments.)

Reviewer comments to Author:
Reviewer: 1

Comments to the Author(s)

The strong point of this manuscript is the systematic variation of the social / contextual embedding of the stimuli to be judged in duration. The same ambiguous stimuli were used with different contextual attributions. Scrambled noise created from recorded street scenes and religious festivals were verbally attributed as stemming from either Hindu or Muslim festivals. The stimuli are ambiguous because the noise is always the same, only the verbal attribution differs. Hindu and Moslem students listened to the sound clips and relatively over-reproduced the sounds of their own (in-group) religious festivals as compared to the other (out-group) religious festival. In a control experiment where the noise was attributed to normal street noise the religious groups did not differ in their reproduction performance. The study seems simple enough but unfolds with different parts to create meaningful complexity. The reproducing is done with only a long (20 s stimulus), with a variation of four different durations (between 1 and 20 s), with the in-group and out-group attribution, or a neutral attribution and an equal share of Muslim and Hindu participants. What is unique here is that participants did not fulfil typical task requirements and stimulus properties which would be similar to everyone (a sinus tone, or even emotionally charged stimuli), but that information on the noise which would trigger an individual group background, a contextual (top-down) embedding significantly influenced the duration reproduction performance in otherwise always identical stimuli.

Here are a few remarks on details in the manuscript:

- When on page 4 the term "ambiguous sound clip" appeared, I did not fully understand what that meant. I had to wait until I was reading the methods section to fully understand it. Since this is an important term in the context of this study I would early on give a fuller explanation so that one can better follow the introduction.

- It is probably not so important but one takes notice and asks why female students were unavailable (page 7) in experiment 1a.
- The power analysis on page 7 leads to $n = 35$ subjects for a two-group comparison. However the decisive interaction effect contains $n = 20$ members of the subgroups.
- In the Results section for all experiments there is the outlier criteria of subjects “erroneously recollecting the source of the sound”. Within the given context I don’t understand this. The participants are given the information of Muslim, Hindu, or neutral context. What is an erroneous recollection of sounds?
- Page 17: Make explicit which group is the ethnic minority in Allahabad, I presume the Muslim community?

Reviewer: 2

Comments to the Author(s)

The study deals with an interesting phenomenon – subjective passage of time related to religious content/attribution in the situation. Own-group attribution is demonstrated to be related to longer time productions of the audio clip. Unfortunately the study remains too superficial and descriptive, not aiming to explain the reported effect. Actually, it also explains but the explanation isn’t totally convincing (as simpler causes of the reported effect aren’t excluded, see below). They also don’t use entirely all means experimental psychology has developed to get closer to the causes of the effect (e.g., subjective reports on how the participants had encoded/reproduced the clips, gradual religiosity related effect of subjective time, within-participants design etc).

Altogether, introduction and experiments themselves doesn’t separate social context from other possibly important aspects in the situations (like emotional meaning, familiarity of the situation/festivals, direction of attention etc) that may capture encoding, perception and emotions, and thus also have an effect on time perception. Used terms like “belongingness” are not very well-defined and may be unnecessarily fancy (especially if we agree that these confusions are there). Please explain these issues more deeply!

Some important experimental details are missing (how many repetitions? how many different clips were used? what is the objective description of them, i.e., how many important details can be read out from there? how was the religiousness of participants taken into account? did the groups differ? what was the "recalling of the source of the sound clip, p10, lines 17-23).

How to separate social context from emotions and other factors? Does top-down influence mean that low-level explanations are ruled out? I don’t think so (as top-down knowledge may easily direct ones’s attention to low-level features in the event and thus exclude or minimize high-level effects; the phenomena like perceptual learning, different memory content or higher perceptual sensitivity may play a role there, that in turn are important in time perception models cited above , p 4 of 22)

Please explain what slope and intercept (p 16 of 22) mean in the context of time perception?

It’s still possible that Exp 1a and 1b reflect individual differences, not only experimental manipulation. I consider it as a weakness that between-participants not within-participants design was used (Exp1a and 1b , Exp2)

Memory content effect and direction of attention effects (relevant to time-perception theories referred to in Introduction) may be relevant here, but aren’t studied more.

Altogether, I suggest major revision so that (1) simpler factors/explanations would be considered; (2) mechanism how religious belongingness could affect time perception (i.e., relate it to used models of time perception); (3) give experimental details, incl independent description of used sound clips (what do they possibly contain for the groups); (4) describe your groups more across dimensions you consider important in your study (religiosity, belongingness, memory span, memory content in the experiment etc) to rule out other and support your conclusions

===PREPARING YOUR MANUSCRIPT===

- one version identifying all the changes that have been made (for instance, in coloured highlight, in bold text, or tracked changes);
- a 'clean' version of the new manuscript that incorporates the changes made, but does not highlight them. This version will be used for typesetting if your manuscript is accepted.

===PREPARING YOUR REVISION IN SCHOLARONE===

- 1) One version identifying all the changes that have been made (for instance, in coloured highlight, in bold text, or tracked changes);
 - 2) A 'clean' version of the new manuscript that incorporates the changes made, but does not highlight them.
 - An individual file of each figure (EPS or print-quality PDF preferred [either format should be produced directly from original creation package], or original software format).
 - An editable file of each table (.doc, .docx, .xls, .xlsx, or .csv).
 - An editable file of all figure and table captions.
- Note: you may upload the figure, table, and caption files in a single Zip folder.
- Any electronic supplementary material (ESM).
 - If you are requesting a discretionary waiver for the article processing charge, the waiver form must be included at this step.
 - If you are providing image files for potential cover images, please upload these at this step, and inform the editorial office you have done so. You must hold the copyright to any image provided.
 - A copy of your point-by-point response to referees and Editors. This will expedite the preparation of your proof.

- Ensure that your data access statement meets the requirements at <https://royalsociety.org/journals/authors/author-guidelines/#data>. You should ensure that you cite the dataset in your reference list. If you have deposited data etc in the Dryad repository, please include both the 'For publication' link and 'For review' link at this stage.
- If you are requesting an article processing charge waiver, you must select the relevant waiver option (if requesting a discretionary waiver, the form should have been uploaded at Step 3 'File upload' above).
- If you have uploaded ESM files, please ensure you follow the guidance at <https://royalsociety.org/journals/authors/author-guidelines/#supplementary-material> to include a suitable title and informative caption. An example of appropriate titling and captioning may be found at https://figshare.com/articles/Table_S2_from_Is_there_a_trade-off_between_peak_performance_and_performance_breadth_across_temperatures_for_aerobic_sc_ope_in_teleost_fishes_/3843624.

Author's Response to Decision Letter for (RSOS-201063.R0)

See Appendix A.

RSOS-201063.R1 (Revision)

Review form: Reviewer 1

Is the manuscript scientifically sound in its present form?

Yes

Are the interpretations and conclusions justified by the results?

Yes

Is the language acceptable?

Yes

Do you have any ethical concerns with this paper?

No

Have you any concerns about statistical analyses in this paper?

No

Recommendation?

Accept as is

Comments to the Author(s)

I had several minor comments which have all been addressed in a satisfactory manner.

Review form: Reviewer 2

Is the manuscript scientifically sound in its present form?

Yes

Are the interpretations and conclusions justified by the results?

Yes

Is the language acceptable?

Yes

Do you have any ethical concerns with this paper?

No

Have you any concerns about statistical analyses in this paper?

No

Recommendation?

Accept as is

Comments to the Author(s)

I'm rather satisfied with the article in its current form. Great idea!

Decision letter (RSOS-201063.R1)

Dear Dr Tewari,

It is a pleasure to accept your manuscript entitled "Group congruent labelling leads to subjective expansion of time" in its current form for publication in Royal Society Open Science. The comments of the reviewer(s) who reviewed your manuscript are included at the foot of this letter.

We have recently revised and improved our policies regarding competing interests and transparency, and we ask that Editorial Board Members ensure to declare their status within the Competing Interests section as follows (and adjust the question to "Yes"):

"At the time of writing, [PROFESSOR NAME HERE] is a Board Member of Royal Society Open Science, but had no involvement in the review or assessment of the paper."

Please therefore ensure that you send to the editorial office a version of your accepted manuscript, with the updated conflict of interest statement at the end of your paper.

You can then expect to receive a proof of your article in the near future. Please contact the editorial office (openscience_proofs@royalsociety.org) and the production office (openscience@royalsociety.org) to let us know if you are likely to be away from e-mail contact -- if you are going to be away, please nominate a co-author (if available) to manage the proofing process, and ensure they are copied into your email to the journal.

on behalf of Dr Alexa Morcom (Associate Editor) and Essi Viding (Subject Editor)
openscience@royalsociety.org

Associate Editor Comments to Author (Dr Alexa Morcom):

Dear Dr Tewari

Thank you for submitting the revised version of your paper manuscript "Group congruent labelling leads to subjective expansion of time" (RSOS-201063) to Royal Society Open Science. Following re-review, I am happy to inform you that we now can accept the paper in its current form. The reviewers and I appreciate your responsiveness to their earlier suggestions. Your paper will make a strong contribution to the literature.

I will now pass your paper to our editorial team who will take it to the production stage. I look forward to seeing it published and to receiving further submissions from you in the future.
Yours sincerely

Dr. Alexa Morcom

Reviewer comments to Author:
Reviewer: 1
Comments to the Author(s)

I had several minor comments which have all been addressed in a satisfactory manner.

Reviewer: 2

Comments to the Author(s)

I'm rather satisfied with the article in its current form. Great idea!

Appendix A

Dear Dr. Morcom,

Many thanks for your decision letter on our manuscript.

We thank you and the reviewers for your thorough engagement with our manuscript. We offer a point-by-point response to your comments and suggestions. We are grateful for your attention.

Response to Editor and Reviewers

Associate Editor 1 (Dr Alexa Morcom)

COMMENT 1

Reviewer 2 questions the theoretical framing of the experiments and asks that you consider simpler explanations that might underpin the effects of the social context manipulation.

RESPONSE

Thanks for the comment. In the present paper, we argue that group congruent/incongruent labelling affect duration perception through changes in the pacemaker component of the internal clock model. Clock model is a commonly used theoretical framework to explain time perception. The Experiment 1a and 1b demonstrates the effect of social context labelling on duration perception for two different religious groups, and the Experiment 2 suggests that this effect might be due to participants paying more attention to the sound clip when it was labelled with ingroup source labelling and this probably affects the pacemaker component of internal clock model (since only slope is influenced). The effect of social/group context manipulation is mediated via simpler cognitive processes such as attention, which in turn changes the pacemaker component in the clock model. We considered simpler factors or explanations (purely bottom-up) and employed control across the three experiments to rule out these factors. Our experimental findings are important for both the social categorisation and time perception literature and has the potential to stimulate further research.

For details, kindly refer to the specific response below (Reviewer 2/Response to Comment 6).

COMMENT 2

Reviewer 1 also asks you to give further detail about the power calculations used to justify the sample sizes. It is essential to justify the samples for the specific effects of primary interest.

RESPONSE

Thanks for the comment. We have given further details and modified the text about power calculations. Kindly refer to the specific response below (Reviewer1/Response to Comment 3).

COMMENT 3

In Experiment 1a, the p value for the post-hoc test for the Muslim groups on p.11 lines 19-20 is quoted as equal to 05. Is this correct, or is it less than .05? At alpha = .05, this distinction is

important as it affects whether this comparison can be declared significant or not, and thus the interpretation of the interaction.

RESPONSE

Thanks for pointing it out. p is less than .05 and it has now been corrected in the manuscript. We have now reported the corrected p value ($p = .04$) (page no. 5)

Reviewer 1

COMMENT 1

When on page 4 the term “ambiguous sound clip” appeared, I did not fully understand what that meant. I had to wait until I was reading the methods section to fully understand it. Since this is an important term in the context of this study I would early on give a fuller explanation so that one can better follow the introduction.

RESPONSE

Thanks. We have now added the following sentence about “ambiguous sound clip” in the first page of the introduction.

“We generated an ambiguous sound clip that contained difficult to identify sounds from various religious festivals and busy city streets mixed with white noise.” (page no. 1)

COMMENT 2

It is probably not so important, but one takes notice and asks why female students were unavailable in experiment 1a.

RESPONSE

Experiment 1a primarily aimed to test effect of source labelling on duration estimation. We were not interested in gender differences at this point. We initially explored the possibility of recruiting equal number of male and female participants from both the religious groups in the sample. Unfortunately, at that point in time, we were not able to attract and recruit many Muslim female students to register for our experiment. This might be partially related to cultural factors. We agree this is not ideal. Therefore, rather than having unequal number of female participants in the two groups (which would raise questions of validity on its own), we thought it would be better to conduct experiment 1a only with male participants. However, in Experiment 2 when we used only Hindu subjects, we have included both male and female participants.

To make it clearer, we have now reworded the following statement in participants details of Experiment 1 as (without mentioning the name of the religious group):

“We restricted our recruitment only to male participants because of the limited availability of female participants in a particular condition.” (Page no. 3)

COMMENT 3

The power analysis on page 7 leads to $n = 35$ subjects for a two-group comparison. However, the decisive interaction effect contains $n = 20$ members of the subgroups.

RESPONSE

We thank the reviewer for pointing out this inconsistency in the way we have reported the sample size. Our sincere apologies. We have now stated the effect for which we computed the effect size and how we have calculated sample size. The effect size was calculated based on a similar social context effect from our earlier study (Srinivasan et al., 2015). We expected a main effect of religious context, which was the effect considered for calculating sample size. The text has been modified as follows:

“A power analysis and a priori sample estimation was performed based on the effect size from Srinivasan et. al. (2015). The effect size (Cohen's d) from Srinivasan et al (2015) was 0.8 for the social context effect. For a between-subjects ANOVA with two factors and two levels in each factor, we expected a main effect of religious context with an effect size of $f = 0.4$ (given $d = 0.8$ from the prior study). With $f = 0.4$, $\alpha = 0.05$ and power = 0.95, we calculated the required sample size of 84. However, expecting some exclusions, we recruited 92 participants.” (Page no. 3-4)

COMMENT 4

In the Results section for all experiments there is the outlier criteria of subjects “erroneously recollecting the source of the sound”. Within the given context I don't understand this. The participants are given the information of Muslim, Hindu, or neutral context. What is an erroneous recollection of sounds?

RESPONSE

The “erroneously recollecting the source of the sound” phrase reflected all those participants who failed to correctly recall the source of sound clip. We have mentioned this in the original manuscript (page number 9 line 17-22). We have now reworded this to make it clearer as follows:

“...because they failed to correctly recall the source of the sound clip after duration reproduction.” (Page no. 4)

COMMENT 5

Page 17: Make explicit which group is the ethnic minority in Allahabad, I presume the Muslim community?

RESPONSE

Yes. Muslims are one of the ethnic minorities in Allahabad (now renamed as Prayagraj) as well as in India. We incorporated this at two places both in introduction and discussion. Now it reads:

“...we presented ambiguous sound clip(s) to participants from two groups from different religions at the study site, Hindus (ethnic majority group) and Muslims (ethnic minority group) ...” (Page no. 3)

“This could be a possible reason for the overall longer duration estimates by the Muslim group (ethnic minority group) compared to the Hindu group (ethnic majority group).” (Page no. 9)

Reviewer 2

COMMENT 1

Altogether, introduction and experiments themselves doesn't separate social context from other possibly important aspects in the situations (like emotional meaning, familiarity of the situation/festivals, direction of attention etc) that may capture encoding, perception and emotions, and thus also have an effect on time perception. Used terms like “belongingness” are not very well-defined and may be unnecessarily fancy (especially if we agree that these confusions are there). Please explain these issues more deeply!

RESPONSE

We agree that many factors like emotional meaning, familiarity, and attention can influence time perception, and we have cited some relevant studies in our manuscript (see second paragraph on page no. 2). We also agree that broader factors like social context might influence time perception via more basic cognitive processes like attention. In our previous study using a similar paradigm and investigating the effect of social context in “Mela” participants, we showed that this effect is mediated via attention during the encoding stage (Srinivasan et al., 2015). We have also cited the above study in our manuscript. The current study extended and tested the similar idea at the group membership level using two religious groups and religious priming. In fact, we have actually tested and explained our results based on the attentional processes in the context of a cognitive model of time perception - ‘the internal clock model’ (see rationale for Experiment 2 in the manuscript, page no. 6).

The present paper provides evidence for the effect of social identity on duration perception an unfamiliar ambiguous sound clip. This is essentially a top-down effect wherein mere group congruent labelling influences duration reproduction. We believe our effect is possibly mediated by attentional allocation based on group identity related processes during the processing of the ambiguous sound clip. The manuscript does include review of studies showing robust effect of social categorisation on social cognitive processing starting from visual perception, joint action task, person perception to political ideology.

How intergroup categorisation is associated with *rich, elaborated and differentiated processing* is discussed in the discussion (page no. 8). Moreover, the audio clips used in the study was unidentifiable controlling for the effects of perceptual attributes of the sensory stimuli in terms of familiarity, emotional association and low-level sensory factors.

However, we agree with reviewers’ comment and discussed it later and discussed the *need for further exploration using an integrative approach to understand the joint effect of perceptual*

attributes of sensory stimuli, the allocation of attentional processes, and the existence of the previously established mental representations related to our categorizations. (Page no. 8)

COMMENT 2

Some important experimental details are missing (how many repetitions? how many different clips were used? what is the objective description of them, i.e., how many important details can be read out from there? how was the religiousness of participants taken into account? did the groups differ? what was the "recalling of the source of the sound clip, p10, lines 17-23).

RESPONSE

Thanks. Information about the details requested are already included in the manuscript but perhaps it was not clear or highlighted. The manuscript contains information on all the above points. For example, description of sound clip is given on page number 4 -*These sounds were jumbled together with white noise to make the sounds unidentifiable. The duration of the sound clip was 20 seconds and the loudness was 85 dB. The 20 second sound clip is available at <https://osf.io/g82jf/>.*"

The manuscript did include the details of trials in Experiment 2 as: *The 20 sec ambiguous sound clip (Experiment 1) was randomly edited into different clips of four durations; 1 sec, 5 sec, 10 sec, and 20 sec. Each sound clip of a specific duration was repeatedly presented for five trials.* (Page no.7)

To bring more clarity on number of trials in experiment 1a & 1b, we added one statement,

"The experiments consisted of a single trial and the identical 20 seconds long audio clip was presented only once in each of the conditions." (Page no. 4)

Participants self-reported their ethnic affiliation. We reworded the statement as

"Participants self-reported themselves as belonging to either Hinduism ..." (page no 3)

Details on other demographic measures is given in response to comment 6.

As mentioned in the original manuscript, before listening to the clip, all the participants were instructed that the sound clip was recorded from a Hindu or Muslim festival or a busy city street. To test whether participants processed the information about the source of the sound and remembered it till the end of the experiment, all participants were asked to recall the source of the sound clip at the end of the experiment (page number 9 line 17-22). "Erroneously recollecting the source of the sound" reflect all those participants who failed to correctly recall the source of sound clip. We reworded "erroneously recollecting the source of the sound" as:

"...because they failed to correctly recall the source of the sound clip after duration reproduction." (Page no. 4)

COMMENT 3

How to separate social context from emotions and other factors? Does top-down influence mean that low-level explanations are ruled out? I don't think so (as top-down knowledge may

easily direct ones' attention to low-level features in the event and thus exclude or minimize high-level effects; the phenomena like perceptual learning, different memory content or higher perceptual sensitivity may play a role there, that in turn are important in time perception models cited above , p 4 of 22)

RESPONSE

As described in the manuscript, we controlled for bottom-up factors by (a) making the sound ambiguous and using the same sound clip(s) in all the conditions and (b) also using a neutral city sound source attribution. Given the actual content of the stimulus is the same in the neutral city sound condition and the two groups do not show any significant difference, this indicates that the *purely* bottom-up sensory processing is the same for both groups. We elaborated and re worded following statements both in introduction and discussion.

“The same ambiguous stimulus was used in all the conditions to control for low-level stimulus factors and to ensure that the effect is due to source labelling.” (Page no. 3)

“This experiment was performed to ensure that there is a difference in duration perception between religious groups only when the labelling is based on religious context (in experiment 1a) and the effect disappears when a common secular label (city sounds) is used.” (Page no. 3)

“Given that the audio clip was identical and ambiguous across all conditions in experiment 1a, the temporal processing changed due to mere in-group or out-group labelling and not to purely bottom-up factors. Different religious groups do not show any difference in duration perception when they were presented with a common secular context (experiment 1b). The results of experiment 2 further confirmed that purely bottom-up sensory processing differences cannot account for our religion-congruent labelling results. The participants interpreted the in-group festival sounds possibly as having meaningful associations resulting in richer encoding of (ambiguous) stimuli. This meaningful association to in-group festivals probably led to top-down influences including motivational processes and attentional processes on mechanisms involved in time perception.” (Page no. 9)

Yes, we agree with the reviewer and it is our major claim that top-down factors do influence the way the information is processed due to attention, memory or other factors. This is precisely what we are arguing for and want to emphasize. More specifically, we argue that simply attributing the source to one's own religious group, makes the subject process the information differently (probably based on differences in attention) leading to differences in reproduced durations.

COMMENT 4

Please explain what slope and intercept (p 16 of 22) mean in the context of time perception?

RESPONSE

The original manuscript did include this information (page 12, lines 42-54 and page 13, lines 2-6).

We have now elaborated the description about slope and intercept:

“In the context of the internal clock model [10], the slope refers to the rate at which the pacemaker encodes the temporal representation, whereas the intercept refers to a constant shift in temporal representation independent of pacemaker and mostly attributed to the switch component. Change in slope implies that the pacemaker component of the model has been possibly influenced whereas changes in intercept (latency) implies that probably the switch component of the clock model has been affected.

Applying this to the group congruent source labelling effect in experiment 1, if group congruent/incongruent labelling leads to changes in the pacemaker component of the internal clock, then the changes in temporal judgments would increase in a multiplicative manner with increase in the magnitude of actual duration, leading to a slope effect [10,12]. On the other hand, if the source labelling leads to changes in the latency of the switch component, then the changes in temporal judgments would increase in an additive manner with increase in the magnitude of actual duration, leading to an intercept effect.” (Page no 6)

COMMENT 5

It’s still possible that Exp 1a and 1b reflect individual differences, not only experimental manipulation. I consider it as a weakness that between-participants not within-participants design was used (Exp1a and 1b, Exp2). Memory content effect and direction of attention effects (relevant to time-perception theories referred to in Introduction) may be relevant here, but aren’t studied more.

RESPONSE

We do appreciate the concern of using a between-subject design compared to a within-subject design. However, we would like to clarify the basis for our choice.

In experiment 1a and 1b, we aimed to compare the effect of group congruent source labelling across the two religious groups. By its nature, religious group is a between subject variable and the group congruent context cannot be common for any of them. We did use a relatively neutral context for both the groups and found no difference in duration estimation. To control for low-level stimulus properties, we used an identical 20 seconds sound clip in both the conditions. This is not possible if we present two different stimuli to the same subjects and attribute two different sources. Also, it would be difficult to employ a within-subjects design by attributing different sources to the identical sound clip. Practice and memory effects would violate the independence of manipulations in a within-participants design. While counter balancing would help, there is a definite possibility of asymmetric order effects depending on which source (in-group or outgroup) is mentioned first. Use of multiple variant sounds would also bring its own complications of content, order, memory, practice effect and so on. Using only one condition per subject ensures that memory of listening once already does not interfere with other conditions. Therefore, to avoid these issues we used a between-subject design in Experiment 1a and 1b.

In Experiment 2, we used a mixed design, as group congruent and incongruent source priming could have potential to order and memory effect. We used within group to test multiple duration. While source labelling is not a within-subject manipulation, random assignment would ensure that there is no selection bias.

However, we appreciate the reviewer's concern regarding individual differences. We minimized the group related individual differences by randomly assigning the group congruent/incongruent attribution conditions across two religious groups. Moreover, we also included a control condition with attribution of common city context, and the two group did not show any significant difference with the same ambiguous stimulus. In experiment 2, we have ensured that subjects are randomly assigned and have checked whether the groups are similar in terms of religiosity and identity.

COMMENT 6

Altogether, I suggest major revision so that (1) simpler factors/explanations would be considered; (2) mechanism how religious belongingness could affect time perception (i.e., relate it to used models of time perception); (3) give experimental details, incl independent description of used sound clips (what do they possibly contain for the groups); (4) describe your groups more across dimensions you consider important in your study (religiosity, belongingness, memory span, memory content in the experiment etc) to rule out other and support your conclusions.

RESPONSE

Thanks for giving opportunity to explain. Some of these are already addressed as mentioned in the responses to previous comments. The manuscript did include following points.

1 - We considered simpler factors or explanations (purely bottom-up) and the controls employed across the experiments we feel rule these factors. We propose that the effect is mediated by better attentional processing (encoding and recall) that changes the pacemaker component in the clock model. We elaborated on this point as mentioned in the response to previous comment.

2 - Experiment 2 aimed to investigate possible mechanism. Using multiple durations, significant slope effect reflects how group congruent source labelling influences pacemaker component in the internal clock model of time perception.

In making our argument (that group congruent labelling affect duration estimation through probably due to changes in pacemaker component of internal clock model), we accept that no single experimental study can address all the issues (theoretical and methodological) involved. However, we believe that these experiments provide initial evidence for this claim. We would also observe that given the significance of the social categorisation to time perception, these results are important and has the potential to stimulate further research. We reworded following statement in the discussion:

“Further studies are needed to elucidate how group congruency influences duration estimation using other models of time perception.” (Page no. 9)

3 - The original manuscript did have some of these details. Further we modified other desired and relevant descriptions as mentioned in the response of previous comments. We hope now the experimental details are clearer.

4 - We thank the reviewer for this. We did compare groups on variables such as religiosity, Hindu identity (belongingness). In the experiment 2, we had collected information on

religiosity, Hindu identity and everyday contact with Hindu/Muslim groups along with few other demographical information at the end of experiment 2 to ensure the two groups (different source labelling) are similar in terms of religious factors. This got missed in the previous version and we added the details now. We did control for religiosity and religious identity in experiment 2.

We have reworded and added the following details in method and results of experiment 2.

“To ensure the groups are similar in terms of religious factors, the participants were asked few questions on the level of religiosity (to what extent you consider yourself as religious?), Hindu identification (To what extent you identify yourself as Hindu?), contact with Hindus (How many Hindu friends do you have?/ How many Hindus live close to your residence?/How often do you spend time with Hindu friends?) and contact with Muslims (How many Muslim friends do you have?/ How many Muslim live close to your residence?/How often do you spend time with Muslim friends?) using five points rating scales (ranging from not at all to a lot) along with demographic details such as age, gender, and education.” (Page no.7)

“We performed t-tests to check whether the two groups differed in terms of religiosity, identity and contact with ingroup and outgroup members. The difference between religiosity for those in the Hindu condition ($M=3.06$, $SD = 0.25$) was not significantly different compared to those in the Muslim condition ($M=2.86$, $SD = 0.26$), $t(37) = .55$, $p = .59$ $d = .18$. Similarly, the difference between Hindu identity across conditions Hindu ($M = 3.56$, $SD = 1.25$) and Muslim condition ($M=3.62$, $SD = 1.20$) was not significant, $t(37) = .16$, $p = .87$, $d = .05$. The two groups were also similar with respect to the rating of everyday contact with Hindus and Muslims. Mean contact ratings with Hindus did not differ significantly between those in the Hindu condition ($M = 4.74$, $SD = 0.39$) compared to those assigned to Muslim condition ($M = 4.84$, $SD = 0.27$), $t(37) = 0.92$, $p = .36$, $d = 0.30$. Similarly groups had similar amount of contact with Muslims across Hindu condition ($M = 2.94$, $SD = 0.87$) and Muslim condition ($M = 2.73$, $SD = 0.65$), $t(37) = 0.88$, $p = .38$, $d = 0.28$.”(Page no. 7)

Based on your suggestion, we took the answer to these questions and used them as covariates in our statistical analyses to ensure that our main findings with social labelling are not affected by these factors. The effects we had reported earlier do not change when we do ANCOVA considering these religion-related measurements as covariates. We have added following in the results.

“To test the effect of group congruent source labelling, we performed a 4 (Duration: 1, 5, 10 & 20 sec) x 2 (source attribution: Hindu and Muslim festivals) mixed group ANCOVA on reproduced durations with religiosity, Hindu identification, contact with Hindu and contact with Muslims as covariates. Greenhouse-Geisser correction was applied as sphericity was found to be violated by Mauchly’s test of sphericity. Significant main effect of context showed longer duration estimates in the Hindu festivals condition ($M = 7.89$, 95% CI: 6.97 sec, 8.81 sec) compared to the Muslim condition ($M = 6.62$, 95% CI: 5.77 sec, 7.47 sec), $F(1, 33) = 7.59$, $p = .009$, $\eta_p^2 = 0.19$. We successfully replicated the group congruent labelling effect that was established in the first experiment. The main effect of duration was also significant $F(1.93, 63.75) = 10.00$, $p < .001$, $\eta_p^2 = 0.23$, showing that all the estimated durations differed significantly from each other (see figure 2). This shows that the participants were able to perform the task and discriminate different durations.

More importantly, the interaction between duration and group congruent/incongruent source attribution was significant, $F(1.93, 63.75) = 7.56$, $p = .001$, $\eta_p^2 = 0.19$. Post-hoc comparisons (Holm-Bonferroni correction) showed longer reproduction for the in-group ($M= 15.86$, 95% CI: 14.06 sec, 17.66 sec) as

compared to out-group condition ($M = 12.67$, 95% CI: 11 sec, 14.33 sec) only for the 20 second duration, $t(37) = 4.90$, $p = .001$, $d = 0.85$. Similar trend was seen in other durations such as 5 second ($M_{\text{Hindu}}=5.42$, 95% CI:4.08 sec, 6.76 sec ; $M_{\text{Muslim}}=4.70$, 95% CI:4.05 sec, 5.34 sec, $t(37)=1.46$, $p=.30$, $d = 0.34$) and 10 second ($M_{\text{Hindu}}=8.75$, 95% CI:7.64 sec, 9.87 sec; $M_{\text{Muslim}}=7.62$, 95% CI:6.62 sec, 8.63 sec, $t(37) = 2.04$, $p = .13$, $d = 0.47$) though the difference was not significant (see Figure 2).

None of the interactions with covariates except between religiosity and durations were significant, $F(1.93, 63.75) = 6.09$, $p = .004$, $\eta_p^2 = 0.16$. This effect of religiosity and duration was independent from the interaction between group congruent source attribution and durations. Further analysis showed that religiosity influences duration estimation at larger durations of 10 and 20 seconds ($ps < .05$) but does not influence duration estimations of 1 and 5 seconds ($ps > .05$).” (Page No. 7-8)

We do not have a clear explanation for this un-hypothesized religiosity x duration interaction, which we plan to follow up with further studies. We have now mentioned this in the discussion section as well and the need for further studies to understand this interaction. It reads:

“The empirical evidence clearly demonstrates state level effect of religious labeling on time perception. The significant association between trait level religiosity and time perception warrants a systematic exploration and explanation in future studies”. (Page no. 10)

We did not measure memory span. We agree with the reviewer that an integrative analysis is required in future investigation.

Conclusion

Thanks again for this opportunity to submit a major revision of this paper. We hope you will agree that we have addressed all the comments and suggestion and the manuscript is thoroughly improved. We also hope you agree that our theoretical point is important and has the potential to be published in your journal.

Kind regards and best wishes

The Authors